# Paclitaxel—A Valuable Tool for Inducing Visceral Pain in Preclinical Testing?

Corina Andrei [iD], Anca Zanfirescu *[iD], Dragoș Paul Mihai [1][iD] and Simona Negreș

Faculty of Pharmacy, "Carol Davila" University of Medicine and Pharmacy, Traian Vuia 6,
020956 Bucharest, Romania
* Correspondence: anca.zanfirescu@umfcd.ro; Tel.: +40-723632617

**Abstract:** Visceral pain is a unique clinical entity that lacks an effective and safe treatment. Proper preclinical models are essential for assessing new drugs developed for the treatment of this pathology. Few studies report that paclitaxel, an antineoplastic agent, can be used to induce visceral pain in laboratory animals. Our purpose was to investigate the reproducibility of these studies and to develop an animal method that would allow assessing consistent visceral pain. In this study, we used four doses of paclitaxel ($3\ mg \times kg^{-1}$; $5\ mg \times kg^{-1}$; $10\ mg \times kg^{-1}$ and $15\ mg \times kg^{-1}$). Visceral pain was evaluated using a scale of abdominal pain immediately after the administration of a single dose of paclitaxel to rats. Tactile and thermal hypersensitivity were assessed using von Frey filaments and the tail flick test initially, at 24 h and 48 h after administration. Rats experienced visceral pain and mechanical and thermal hypersensitivity 24 h after the administration of paclitaxel. The intensity of the pain was increased in a dose-dependent manner with the highest intensity of effect being observed after the administration of a dose of $15\ mg \times kg^{-1}$. Paclitaxel induces visceral pain. The dosage varies depending on the employed strain of rat. This method allows for assessing the efficacy of analgesics to be useful against visceral pain if the paclitaxel dose is adjusted accordingly to the animal strain.

**Keywords:** paclitaxel; visceral pain; tactile hypersensitivity; thermal hypersensitivity; scale of abdominal pain

## 1. Introduction

Visceral pain results from the activation of the sensory spinal afferents that innervate internal organs and is one of the most frequent types of ache [1,2]. Visceral pain is mostly diffuse, since pain localization and intensity can often vary throughout the body [3–5]. There is no direct correlation between the pain's intensity and the severity of the underlying disease. Viral gastroenteritis can lead to severe abdominal pain, while colon cancer particularly in initial stages, generates low-intensity pain. In addition, harmless stimuli can cause increased pain when acting on inflamed tissue [2].

Most frequently, visceral pain is associated with functional gastrointestinal disorders, such as irritable bowel syndrome, a disease affecting 10–15% of Europe and U.S. populations with costs exceeding 40 billion, and gynecological disorders, e.g., severe pelvic pain [6,7]. Other forms of visceral pain include dysmenorrhea, renal colic, etc. [6,8,9].

Unlike somatic pain, the mechanism of visceral pain is incompletely elucidated, partly due to the different and complex functions of the viscera, partly due to its complex etiology. Both central and peripheral mechanisms contribute to visceral pain. Central mechanisms include the hyperexcitability of ascending spinal neurons (central sensitization) and the dysregulation of descending pathways that modulate spinal nociceptive transmission [10]. The peripheral mechanisms underlying persistent hyperalgesia include enhanced sensitivity to normal intraluminal contents, nerve injury (sensitization of primary sensory afferent innervating the viscera or nerve damage, e.g., neuropathy) and alteration of various ligand- and voltage-gated channels in sensory neurons [11].

This clinical entity is associated with great psychological stress, cognitive impairment, sleep disturbances and a significant decrease in productivity and the patient's life quality [6,12,13]. Subsequently, it poses a high socioeconomic burden [14].

Furthermore, analgesics have limited the efficacy in treating visceral pain [14], and their usage is limited by specific side effects [14]. Opioids, nonsteroidal anti-inflammatory drugs (NSAIDs) and paracetamol are the main lines of treatment for visceral pain, depending on its intensity [10,15,16].

Therapeutic alternatives include gabapentin, tricyclic antidepressants or serotonin and norepinephrine reuptake inhibitors that can be used in combination with opioids [17–20]. However, the evidence from clinical trials supporting their use is weak [21]. The lack of adequate treatment is a major contributor to the burden associated with this type of pain.

Novel analgesics must be discovered to address this health issue. Thus, the development of animal models for specifically assessing visceral pain is essential for both identifying the underlying mechanisms of this pathology and the screening of new, effective analgesics with potential use in treating this type of pain.

Most animal models of visceral pain are based on the intraperitoneal administration of a chemical stimulus which induces sensitization of the peripheral and central pain pathways. Acetic acid, 2,4,6-trinitrobenzene sulfonic acid, zymosan and cyclophosphamide are the most frequently used, inducing symptoms in animals similar to those seen in patients suffering from this condition [22–24]. One major limitation of using these chemical stimuli is the lack of reproducibility in the biological response [25]. Although animal models based on the administration of a chemical stimulus are widely used to determine the analgesic effect of various substances, the major disadvantage of these models is represented by their lack of specificity. A positive response is not always correlated with an analgesic effect [26]. For example, substances that affect the animal's motor performance may produce false positive responses [27].

The antineoplastic drug paclitaxel is used to treat people with breast, ovarian and lung cancer, the active substance being first isolated from Pacific yew (*Taxus brevifolia*) [28–31]. In cancer cells, paclitaxel targets the beta-tubulin subunit in microtubules and prevents cell division by promoting microtubule stability, thus inhibiting depolymerization. Therefore, paclitaxel induces cell arrest in the G2/M-phase of the cell cycle, leading to cell replication inhibition and apoptosis. However, the pro-apoptotic effect of paclitaxel seems to occur by modulating gene transcription, including DNA-damage response proteins, cytokines and other proteins with key roles in controlling inflammation, proliferation and apoptosis [32].

The pharmaco-toxicological profile of paclitaxel is characterized by various side effects, such as hair loss, allergic reactions, diarrhea, bone marrow suppression and lung inflammation. Moreover, this drug causes chronic and acute hypersensitivity [33–35]. Acute hypersensitivity is felt by the patient as a high-intensity visceral or somatic pain, while chronic hypersensitivity is related to peripheral neuropathy [36,37]. According to data from the literature, in these patients, visceral pain is also associated with somatic hypersensitivity to nociceptive stimuli [38]. Furthermore, a relatively recent study reported a case of acute abdomen-associated visceral pain caused by paclitaxel-induced bowel perforation. The 79-year-old patient suffered from bowel wall ischemia and necrosis, discovered 2 weeks following paclitaxel infusions [39]. Moreover, the necrotizing enterocolitis secondary to taxane chemotherapy is caused by direct mucosal injury and neutropenia-related impaired immune defense, leading to the predisposition of the bowel to infections. Common symptoms associated with chemotherapy-induced necrotizing enterocolitis include abdominal pain and distension, blood or mucus in the stool, and diarrhea and fever [40].

Based on these findings, paclitaxel has often been used to develop animal models of neuropathic pain, however, reports on its use for induction of visceral pain are scarce. Although paclitaxel induces visceral pain in humans by promoting bowel perforation, the underlying mechanism responsible for paclitaxel-induced acute visceral pain in laboratory animals is not fully understood. According to one study, paclitaxel has been investigated for the induction of visceral pain in Wistar rats [41]. The doses used in the study varied

between 0.3 and 3 mg × kg$^{-1}$ [41]. The following parameters were investigated: visceral nociception, burrowing behavior, mechanical nociceptive threshold, heat nociceptive latency, and chemical hypersensitivity. The authors of the aforementioned study suggested that the TRPV1 calcium channel participates in paclitaxel-induced acute visceral nociception since the administration of a TRPV1 antagonist, or TRPV1-positive sensory fiber ablation, significantly reduced paclitaxel-induced spontaneous acute visceral nociception, mechanical allodynia and thermal hypersensitivity [41].

Therefore, the present study was designed to investigate the reproducibility of previous studies and to understand if paclitaxel is suitable for developing an animal method that would allow a consistent assessment of visceral pain-related behavior and the effectiveness of analgesics that would especially address this type of pain. We used the abdominal pain scale to qualitatively assess visceral nociception immediately after the administration of paclitaxel and the associated tactile (using von Frey test) and thermal (using the tail flick test) hypersensitivities as extensions of visceral nociception at 24 and 48 h after administration.

## 2. Materials and Methods

### 2.1. Animals and Treatments

Experimental procedures were performed in accordance with bioethics norms proposed by the NIH Guide for the Care and Use of Laboratory Animals. The experimental protocol was approved by the Bioethics Committee of the Faculty of Pharmacy, University of Medicine and Pharmacy Carol Davila, Bucharest, Romania (CFF05/01.04.2020), and all procedures complied with the ARRIVE guidelines.

Female Wistar rats (10–12 weeks old) were acquired from INCDMI Cantacuzino (Cantacuzino National Institute of Research, Bucharest, Romania) and were maintained in plexiglass cages at a 12 h/12 h light-dark cycle, under constant humidity (35–45%) and temperature (20–22 °C), monitored with a thermohygrometer. Food (rodent ground chow, INCDMI Cantacuzino, Bucharest, Romania) and drinking water were provided ad libitum to the animals.

Rats were left to acclimatize with the new habitat for one week, and thereafter were divided into five experimental groups. Body weights were measured every two days. The treatment was administered as a single intraperitoneal dose as follows. Group C (control group): distilled water 1 mL × kg$^{-1}$; group PAC1: paclitaxel 3 mg × kg$^{-1}$; group PAC2: paclitaxel 5 mg × kg$^{-1}$; group PAC3: paclitaxel 10 mg × kg$^{-1}$; and group PAC4: paclitaxel 15 mg × kg$^{-1}$.

We used as a starting point the highest dose used in paclitaxel-induced visceral pain studies [41].

For this study, we used 40 rats divided in 5 equal groups (8 animals per group). The sample size was calculated based on the amount of variability between the experimental groups. Variability was determined using data collected from a preliminary experiment carried out under identical conditions to the planned experiment (data not shown) [42].

The paclitaxel solution was prepared by diluting the 6 mg/mL solution (Actavis, Romania) with saline to concentrations of 0.3 mg/mL; 0.5 mg/mL; 1 mg/mL and 1.5 mg/mL. The solutions were stored in the refrigerator (2–8 °C) for a maximum of two days.

### 2.2. Tests for the Evaluation of Visceral Nociception

Visceral nociception was qualitatively evaluated using a scale of abdominal pain immediately after paclitaxel administration. Tactile and thermal hypersensitivity induced by paclitaxel were determined before administration (baseline sensitivity) at 24 h (day 3) and 48 h (day 4) after the administration (Figure 1).

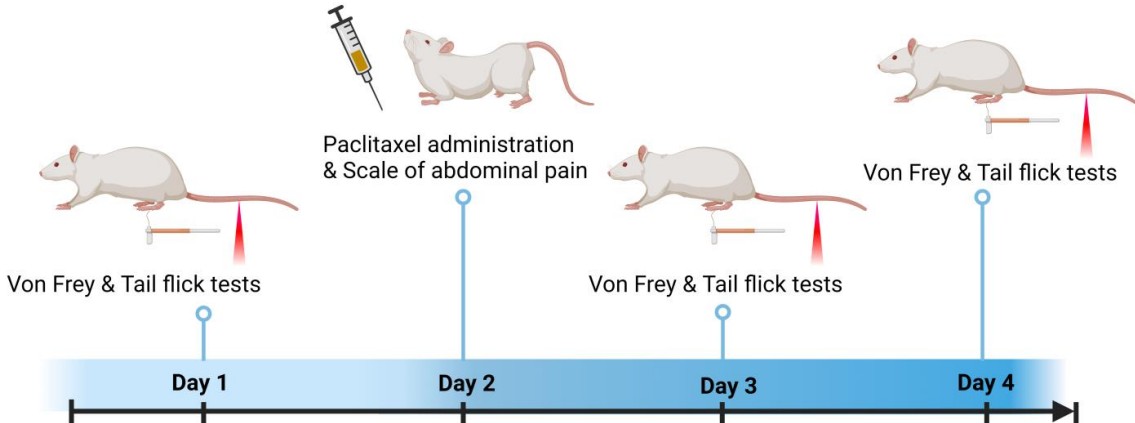

**Figure 1.** Overview of paclitaxel administration and tests for the evaluation of visceral pain and associated thermal and mechanical hypersensitivity.

### 2.2.1. Scale of Abdominal Pain

Qualitative assessment of visceral pain was determined the first 60 min after the intraperitoneal administration of saline and paclitaxel using an abdominal pain scale with a score from 0 to 3. The animals were placed in individual plexiglass observation cubicles and observed for 60 min. A score was established for every 5 min. The mean score was calculated. Scores 0 to 3 were given as follows:

Score 0—normal position of the body and exploratory behavior.

Score 1—leaning position on either side of the body.

Score 2—stretching of the hind limbs, dorsiflexion of the hind paws and body stretched and flat on the bottom, frequently with the pelvis rotated sideward.

Score 3—contraction of the abdominal muscles followed by the extension of the body and the hind limbs [41,43].

### 2.2.2. Tactile Hypersensitivity

Tactile hypersensitivity was evaluated by measuring the withdrawal thresholds of the hind paw using von Frey filaments (Ugo Basile, Gemonio, VA, Italy). The rats were placed in individual plexiglass cages above a perforated wire platform that allowed full access to the plantar areas of the hind limbs. Animals were left to acclimatize for 30 min. Von Frey filaments with increasing stiffness were applied perpendicular to the plantar surface of both hind paws of animals with sufficient pressure to bend the filaments for 6 s. The forces (g) applied were: 1; 1.4; 2; 4; 6; 8; 10 and 15 g (corresponding to the sizes: 4.08; 4.17; 4.31; 4.56; 4.74; 4.93; 5.07; 5.18). The filaments were chosen so that the last filament with the highest strength did not exceed 10% of the weight of the rats, and the test was initiated with the fourth filament of the series (with a force of 4 g). The absence of the paw withdrawal was considered a negative response (marked with O), and in this case, the following filament with higher stiffness was used. Paw retraction was correlated with a positive response (marked with X) and led to the use of the next filament with less stiffness. A total of 6 responses were determined from obtaining an OX or XO sequence or 4 consecutive positive or negative responses. The 50% response threshold was calculated using the Dixon up-and-down method [44,45], applied and validated by Chaplan et al. [46].

### 2.2.3. Thermal Hypersensitivity

Thermal hypersensitivity was assessed using the tail flick test (Ugo Basile, Gemonio, VA, Italy). The assessed parameter was the tail-flick latency [47]. The thermal stimulus (a high-intensity light beam), applied to the rat's tail, induces a painful sensation of heat, followed by a tail-withdraw reflex. An intensity of 50% was maintained throughout the experiment. The device automatically recorded the latency time. A maximum latency of 10 s was set to prevent tissue damage [48].

### 2.3. Statistical Analysis

Statistical analysis and graphical illustration of the experimental data were performed using GraphPad Prism software package v.5.00 (GraphPad Software, San Diego, CA, USA). The type of data distribution was determined using the D'Agostino–Pearson test. The experimental results were analyzed using the following statistical tests: the one-factor Analysis of Variance (ANOVA) test followed by the Bonferroni post hoc test for parametric data and the Kruskal–Wallis test followed by the Dunn post hoc test for nonparametric data. The level of statistical significance was $\alpha = 0.05$, the confidence interval was 95% and the experimental results were expressed as individual mean values $\pm$ standard error of the mean (S.E.M.).

### 3. Results

#### 3.1. Assessment of Visceral Pain Using the Scale of Abdominal Pain

Abdominal pain reaction scores were significantly impacted by the paclitaxel administration. Changes in the behavior of the experimental animals were observed depending on the administered dose (univariate ANOVA, $F(4,35) = 32.3$, $p < 0.0001$, Figure 2A). The most pronounced effect was observed at group PAC4—15 mg $\times$ kg$^{-1}$ ($p < 0.0001$, Bonferroni correction). The effects of paclitaxel in group PAC2—5 mg $\times$ kg$^{-1}$ and PAC3—10 mg $\times$ kg$^{-1}$ were relatively similar ($p < 0.01$, Bonferroni correction). The lowest algogenic effect was observed at group PAC1—3 mg $\times$ kg$^{-1}$ ($p > 0.05$, Bonferroni correction). A 3.70-fold increase in the pain reaction score was noticed for group PAC1 when compared to the lowest administered dose (group PAC1), while for groups PAC3 and PAC2, the scores were increased by 0.700-fold and 0.470-fold, respectively. Acute pain was observed in laboratory animals approximately 5–10 min after administration with a maximum intensity of 20–30 min and lasted 40 min for the highest dose of paclitaxel used in the experiment (Figure 2B).

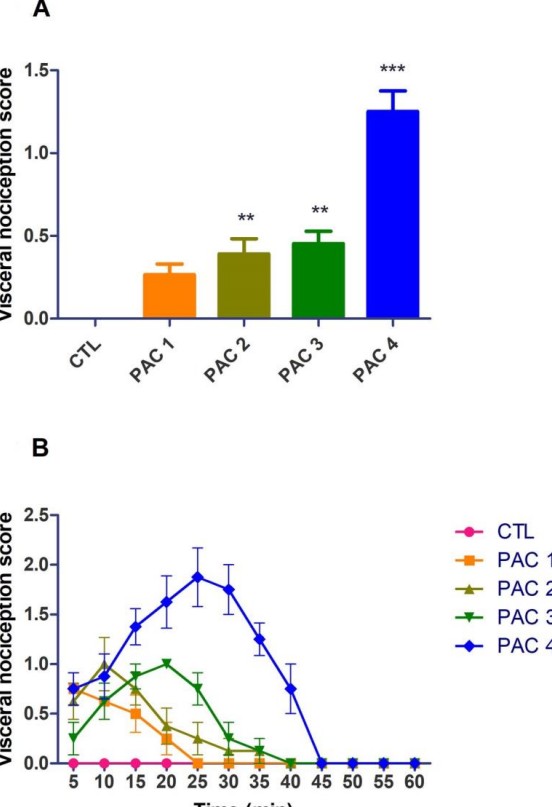

**Figure 2.** Evaluation of visceral pain. (**A**) Scale of abdominal pain—variation of visceral nociception score between treated groups. Data are presented as mean of visceral nociception score $\pm$S.E.M.

CTL—control (n = 8); PAC1—paclitaxel 3 mg × kg$^{-1}$ (n = 8); PAC2—paclitaxel 5 mg × kg$^{-1}$ (n = 8); PAC3—paclitaxel 10 mg × kg$^{-1}$ (n = 8); PAC4—paclitaxel—15 mg × kg$^{-1}$ (n = 8). ** $p < 0.01$ vs. CTL; *** $p < 0.001$ vs. CTL. (**B**) Evolution of abdominal pain over time after administration of a single dose of paclitaxel. Data are presented as mean ± S.E.M. of visceral nociception score. CTL—control (n = 8); PAC1—paclitaxel 3 mg × kg$^{-1}$ (n = 8); PAC2—paclitaxel 5 mg × kg$^{-1}$ (n = 8); PAC3—paclitaxel 10 mg × kg$^{-1}$ (n = 8); PAC4—paclitaxel 15 mg × kg$^{-1}$ (n = 8).

### 3.2. Tactile Hypersensitivity

The influence of a single intraperitoneal paclitaxel dose on the tactile hypersensitivity was assessed 24 h and 48 h after the drug treatment.

An initial test was performed before the administration of the substance and no significant differences were observed. The groups were homogeneous on terms of tactile sensitivity (Figure 3). Additionally, no significant differences in tactile sensitivity were observed for the control group (without treatment) throughout the experiment.

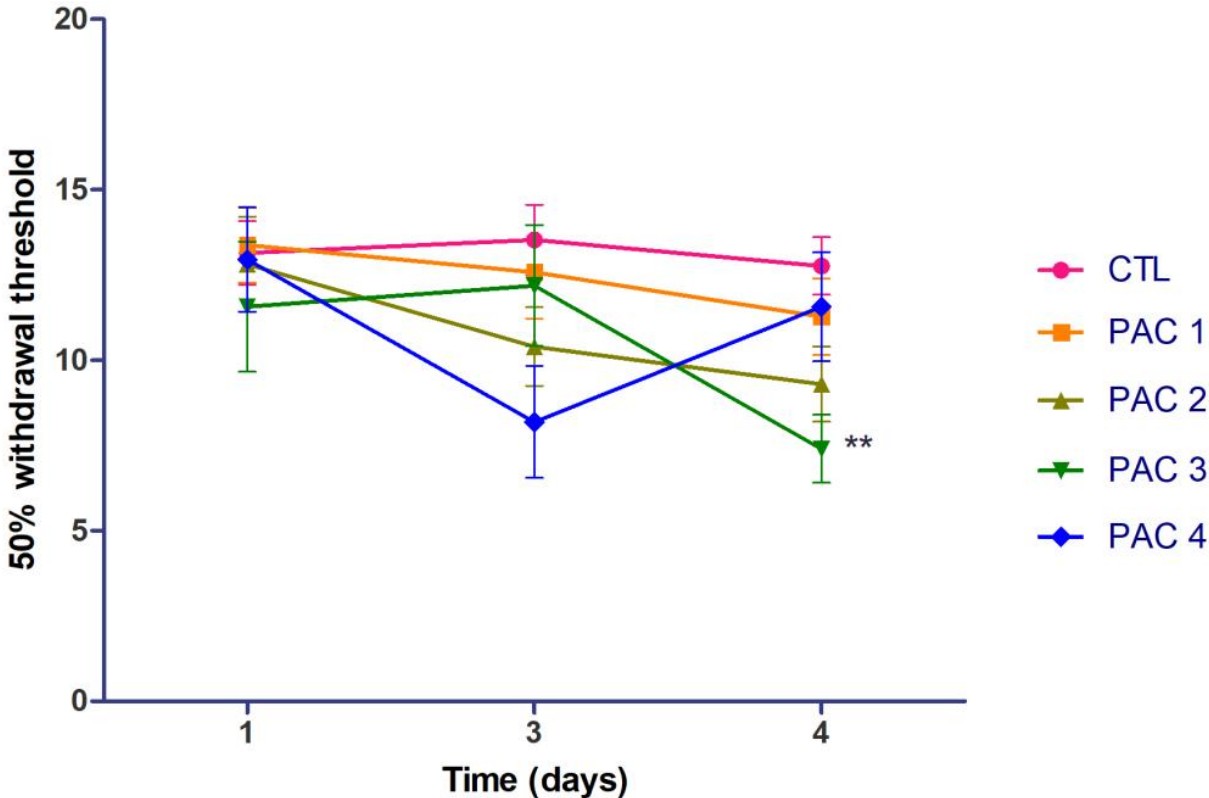

**Figure 3.** Time—dependent variation of tactile hypersensitivity after administration of a single dose of paclitaxel over time. Data are presented as mean ± S.E.M. of 50% withdrawal threshold. ** $p < 0.01$ vs. CTL. CTL—control (n = 8); PAC1—paclitaxel 3 mg × kg$^{-1}$ (n = 8); PAC2—paclitaxel 5 mg × kg$^{-1}$ (n = 8); PAC3—paclitaxel 10 mg × kg$^{-1}$ (n = 8); PAC4—paclitaxel—15 mg × kg$^{-1}$ (n = 8).

Although a reduction of the 50% withdrawal threshold was noticed for all doses compared to the control group, the changes of the pain responses were insignificant after 24 h of administration (univariate ANOVA, $F_{(4,35)} = 2.21$, $p = 0.0882$, Figure 3).

However, a significant effect was observed after 48 h (univariate ANOVA, $F_{(4,35)} = 3.34$, $p = 0.0203$, Figure 3).

There was a statistically significant decrease of the 50% withdrawal threshold for group PAC3 (10 mg × kg$^{-1}$) ($p < 0.01$, Bonferroni correction) compared to the control group

(CTL), and for the other groups, no significant changes were noticed compared to the CTL group, after 48 h.

However, following the assessment of tactile sensitivity after paclitaxel injection at a dose of 15 mg × kg$^{-1}$, we noticed a decrease in the response by 39.5% on day 3 and by 9.39% on day 4, while paclitaxel 1 mg × kg$^{-1}$ lowered the threshold by 9.90% after 24 h and by 42.04% after 48 h, compared to the control group. A decrease of 23.1%, when compared to the control group, was recorded for group PAC2 at 24 h after paclitaxel treatment, and on day 4, there was a 27.2% decrease, but without statistical significance ($p > 0.05$).

### 3.3. Thermal Hypersensitivity

No significant variations were observed in thermal hypersensitivity after paclitaxel administration when compared to the control group on day 3 (univariate ANOVA, $F_{(4,35)} = 0.350$, $p = 0.842$) and 4 (univariate ANOVA, $F_{(4,35)} = 1.43$, $p = 0.244$, Figure 4).

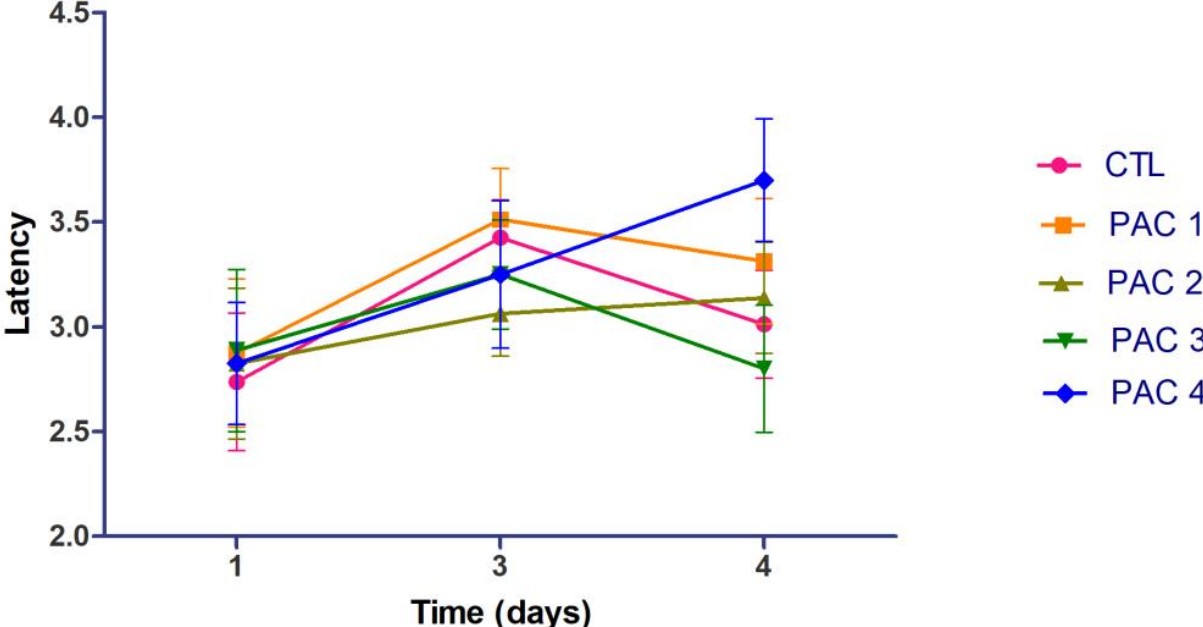

**Figure 4.** Time—dependent variation of thermal hypersensitivity after administration of a single dose of paclitaxel over time. Data are presented as mean ± S.E.M. of latency. CTL—control (n = 8); PAC1—paclitaxel 3 mg × kg$^{-1}$ (n = 8); PAC2—paclitaxel 5 mg × kg$^{-1}$ (n = 8); PAC3—paclitaxel 10 mg × kg$^{-1}$ (n = 8); PAC4—paclitaxel—15 mg × kg$^{-1}$ (n = 8).

On day 3, the reaction time was lowered by 5.1% in groups PAC4 and PAC3 and by 10.6% in PAC2 when compared to the control ($p > 0.05$). The lowest reaction time was recorded on day 3 for the lowest dose of paclitaxel ($p > 0.05$). On day 4, only the dosage of 10 mg × kg$^{-1}$ paclitaxel reduced the tail withdrawal latency, when compared to the control group, by 7.06% ($p > 0.05$).

### 4. Discussion

In clinical practice, abdominal pain frequently occurs after intraperitoneal administration of paclitaxel in patients with ovarian, gastric and pancreatic cancer [49,50]. The abdominal pain felt by most patients is moderate and lasts more than 72 h [50].

In this study, we observed acute pain immediately following intraperitoneal administration of a single dose of paclitaxel to rats [49,51]. The pain was quantified using the abdominal pain scale. An increased score was associated with the presence of visceral nociception. The intensity of pain felt by rats was directly proportional with the increase in dose. The duration of visceral pain in our study was approximately 40 min. Abdominal nociception after intraperitoneal paclitaxel injection to rats was observed in another study,

but the duration of pain was 100 min [41]. This difference might be the consequence of using different doses of paclitaxel. The most intense acute pain was observed in Wistar rats treated with the highest dose of paclitaxel [41]. Although we used different doses of paclitaxel compared to the doses used in the literature, the maximal effect was also observed in our case for the highest administered dose. Although a difference was observed between the duration of pain in animals, the intensity of pain correlated with the dose of paclitaxel supports the idea that the method is reproducible, offering consistent results. However, the dosage that induces visceral pain may vary depending on the rat strain.

The duration of pain observed in rats after administration of the drug was shorter compared to the pharmacodynamic parameter recorded in humans (7 days). Most likely, the discrepancy between the duration of acute pain in humans and laboratory animals occurs due to species specific differences [52].

Additionally, rats showed side effects following the single-dose administration of paclitaxel. The presence of adverse reactions was correlated with the administered dose. For the highest dose used in our preclinical study, we observed severe discomfort especially at the site of administration. Signs and symptoms of discomfort were represented by licking, grooming of the administration site, apathy and shaking of the hind paw. The length of the observed behaviors was longer for the higher dose of paclitaxel compared to the other doses (2 days after administration).

Further, we investigated if paclitaxel-induced visceral pain is followed by mechanical and thermal hypersensitivity. Based on the idea that inflammatory agents can activate nociceptors connected to unmyelinated C and thinly myelinated A$\delta$ fibers, which are responsible for hypersensitivity, we chose to determine thermal and mechanical hypersensitivity after a single dose of paclitaxel [23]. Additionally, paclitaxel-associated acute pain syndrome is an important adverse reaction described by approximately 58% of patients [53]. It occurs 1–3 days after administration of the chemotherapeutic drug and lasts about 7 days in some cases [37,52,53]. It has been hypothesized that paclitaxel-associated acute syndrome is a result of sensitization of nociceptors, their fibers or the spinothalamic system [36,53,54]. Thus, we evaluated the mechanical hypersensitivity 24 h and 48 h after administration of the substance using von Frey filaments. Following administration of a single dose of paclitaxel, rats experienced tactile hypersensitivity. The intensity of tactile hypersensitivity perceived by laboratory animals after a 15 mg $\times$ kg$^{-1}$ dose of paclitaxel increases over time. For the other three groups, a decrease in mechanical hypersensitivity can be observed over time. Not all results were statistically significant, but a reduction in the 50% withdrawal threshold was observed for all doses of paclitaxel 24 h after intraperitoneal administration to rats. This effect has been highlighted in previous studies. Therefore, it can be stated that the occurrence of mechanical nociception 24 h after paclitaxel administration is in line with previous studies [41]. The method of inducing visceral pain with paclitaxel is reproducible, based on similar effects observed in laboratory animals after the administration of the active substance.

To assess the potential induction of thermal hypersensitivity, we used the tail flick test, which involves the application of a heat stimulus, such as radiant heat, to the tail of the rat [48]. Although the results were not statistically significant, a reduction in reaction time was observed 24 h after the administration of the chemotherapeutic agent, compared to the control group. A correlation exists between the dose of paclitaxel used and the thermal hypersensitivity observed in the animals. However, on day 4, for most of the study groups, its effect was unnoticeable.

One limitation of our study is represented by the inconsistent peripheral hypersensitivity observed in rats. Another disadvantage is the necessity of high doses of paclitaxel. On the other hand, the paclitaxel-based animal model produced significant visceral pain behavior in rats, which was consistent with other studies. The associated tactile and thermal sensitivity could be further investigated in future experiments by assessing peripheral hypersensitivity at different time points and in comparison with other, better established models of visceral pain.

Acute nociception followed by hypersensitivity after paclitaxel administration is common in both laboratory animals and humans. The similar behavior may be owed to the specific mechanism of action of paclitaxel (as opposed to chemicals currently used to induce visceral pain). Paclitaxel-associated neuropathic pain is associated with the activation of transient receptor potential vanilloid 1 (TRPV1) [41,55–59]. TRPV1 is a non-selective cation channel with high permeability for $Ca^{2+}$, expressed preferentially in visceral afferents. TRPV1 is activated by capsaicin, noxious heat, acid, mediators of inflammation and ischemia such as protons or lipoxygenase products and mediate thermal and chemical pain [55,60–62]. Future research might focus on investigating the role of TRPV1 receptors in the development of visceral pain induced by paclitaxel.

Paclitaxel also activates toll-like receptor 4 and is involved in the synthesis and release of proinflammatory cytokines (TNF-$\alpha$ and IL-1$\beta$) [52,63,64]. The toll-like receptor 4 (TLR4), activated by endogenous and exogenous ligands, is able to activate microglia, which is important for visceral pain [65]. The role of the TLR4 in visceral pain modulation is supported by a preclinical study that showed a decrease in the intensity of visceral pain in laboratory animals after the administration of the antagonist of this receptor (TAK-242) [66]. Moreover, an increase in peripheral TLR4 activity and proinflammatory cytokine levels was observed in an animal model of visceral hypersensitivity induced by exposure to a high-fat diet [67].

There are some differences between visceral pain induced by intraperitoneal administration of paclitaxel and paclitaxel-induced neuropathic pain. First, for the induction of visceral pain, rats receive a single dose of paclitaxel [41], whereas multiple doses (four doses) are required for neuropathic pain [68,69]. For induction of neuropathic pain, four doses of paclitaxel of 2 mg/kg body are administered, a total of 8 mg/kg [69], and for visceral pain, paclitaxel was administered in doses ranging from 3 to 15 mg/kg. Following the evaluation of thermal and mechanical hypersensitivity in the two animal models of pain, a more pronounced effect was observed for rodents receiving the treatment regimen for neuropathy induction [68,69]. At least two similarities can be identified between neuropathic and nociceptive pain induced by paclitaxel. Although the mechanisms underlying the occurrence of pain are incompletely elucidated, it has been observed that the TRPV1 receptor is involved in the occurrence of both types of pain [41]. Moreover, both types of pain may limit the use of this chemotherapeutic agent.

**5. Conclusions**

In this study, we observed that acute intraperitoneal administration of paclitaxel in rats induced significant visceral pain followed by, to a lesser extent, mechanical and thermal hypersensitivity in a dose-dependent manner. Considering our observations, this animal model can be used to evaluate the efficacy of various analgesics against visceral pain. Further studies are warranted to fully understand the underlying molecular pathophysiological mechanisms involved in taxane-induced visceral pain.

**Author Contributions:** Conceptualization, S.N.; Data curation, C.A. and D.P.M.; Formal analysis, D.P.M.; Investigation, C.A. and A.Z.; Methodology, S.N.; Writing—original draft, C.A.; Writing—review & editing, A.Z. and S.N. All authors have read and agreed to the published version of the manuscript.

**Funding:** This work was co-financed by the European Social Fund, through the Operational Program Human Capital, project number POCU/993/6/13/154722.

**Institutional Review Board Statement:** The animal study protocol was approved by the Bioethics Committee of the Faculty of Pharmacy, University of Medicine and Pharmacy Carol Davila, Bucharest, Romania (CFF05/01.04.2020), and all procedures complied with the ARRIVE guidelines.

**Informed Consent Statement:** Not applicable.

**Data Availability Statement:** All data generated or analyzed during this study are included in this published article.

**Conflicts of Interest:** The authors declare no conflict of interest.

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
