# Peer review of "Paclitaxel—A Valuable Tool for Inducing Visceral Pain in Preclinical Testing?"

_2673-8937, doi:10.3390/ijtm3010010_

Round 1

Reviewer 1 Report

In this study, Andrei et al. used different doses of paclitaxel to induce visceral pain. The idea is interesting, however, more information should be provided. 

1.Does paclitaxel injected patients also suffer from the visceral pain? Although authors mentioned one study (ref. 22), more detailed information should be mentioned in the introduction part. 

2.  How can visceral pain could be differentiated from paclitaxel-induced neuropathic pain? What characteristics of visceral pain could be obtained by using paclitaxel? What is the pro- and cons-?

3. The highest dose (15 mg/kg) used in the study appears to be high as the clinical doses is approximately 8 mg/kg. The rats did not show any side effects?

4. Paclitaxel-induced mechanical and thermal pain is known to be induced at least several days following the injection. Why the authors assessed only in the first 4 days? in 4 days, it is normal that the pain will not be shown. 

5. Maybe it will be worth to assess in the first few hours after the injection just to compare with the visceral pain induced in rats. 

Author Response

Dear reviewer,

Thank you for your assessment of our paper, your questions raised valid concerns and we tried to address them to the best of our capabilities. We hope our replies clarify the issues raised.

Reviewer 2 Report

The manuscript entitled " Paclitaxel – a valuable tool for Inducing Visceral pain in pre- 2 clinical testing?" contains fascinating facts about visceral pain and is well written. The pathway-related figures are well organized, and the analyses of the collected data have been presented systematically. The study is crucial for visceral pain management. The following points, however, are addressed:

Abstract: Please rectify the significant grammatical mistake found in the abstract.

Results:

  1. Please revise the description of "Tactile hypersensitivity" that is found on lines 186-214 since it is difficult to understand.
  2. The single mention of the term "Figure 4" in Line 218 is sufficient for both findings.

Discussion: Only the highest dosages are necessary (lines 242 and 244), therefore there is no need to include the amounts in brackets. It is also not essential to cite "Figure 3" and "Figure 4" (Lines 264 and 276) in this manner since they have already been mentioned in the result sections.

Conclusions: Please rewrite the conclusion so that it focuses only on the results of the experiments as they relate to the specifics of your research question.

Author Response

(The authors gave the same response as above.)

Reviewer 3 Report

1. English writing needs to be checked.
2. The number of investigated animals in each group (sample size) is not mentioned in the relevant section (Animals and treatments).
3. This study was conducted on laboratory animals and must have received approval and code from the ethics committee before starting, but the approval and code of ethics were not written in the manuscript.
4. In line 107, it is written that the original drug solution was 6 mg/mL, so how was 3 mg/mL prepared from it? Concentration units should be checked carefully.
5. The manuscript should be updated with new references (2020-2023).
6. Lines 64 and 65 have no reference. There are useful references about Paklitaxol that are recommended to be visited, for example:
https://doi.org/10.1016/j.ijpharm.2017.05.016
https://doi.org/10.1080/10408347.2017.1416283
https://doi.org/10.1016/S1875-5364(20)60032-2
10.17305/https://doi.org/bjbms.2016.674

Author Response

(The authors gave the same response as above.)

Round 2

Reviewer 1 Report

The authors have clearly answered to all the points.